# A Comparative Study on Planning Patterns of Industrial Bases in Northeast China Based on Spatial Syntax

**Rui Han** [1,2,3,*] **, Daping Liu** [2] **, Guangjie Zhu** [1] **and Linjie Li** [1]

[1] College of Art and Design, Creative Center for ArtSciArch, Jilin Jianzhu University, Changchun 130118, China; Zhugj@sina.com (G.Z.); LiLi456@gmail.com (L.L.)

[2] School of Architecture, Key Laboratory of Cold Region Urban and Rural Human Settlement Environment Science and Technology of Ministry of Industry and Information Technology, Harbin Institute of Technology, Harbin 150006, China; ldp_abc@hit.edu.cn

[3] Department of Architecture and Design, Politecnico di Torino, 10125 Torino, Italy

\* Correspondence: archanrui@sina.com; Tel.: +86-186-431-999-88

**Abstract:** After World War II, unprecedented and positive industrialization and urban construction were launched in lots of developing countries all over the world. Meanwhile, more and more far-reaching planning theories and technological achievements emerged. In this study, we combed the development process of the industrial base planning pattern created by the Soviet Union in the 1950s, summarized its main theoretical and technical contents and its transfer to Northeast China, and revealed the absorption and innovation of this planning pattern in three import industrial cities built in the 1950s in Northeast China. Based on the spatial syntax theory and technology, the practice of three representative industrial bases' planning patterns was deeply analyzed. A comparative study on the theoretical and technical fit planning level among the three bases was implemented from the two aspects of the extension of different functional spatial modules and the connection and accessibility of the road axis. It was finally found that the planning pattern of the new industrial base of a single plant had more advantages of functional support and road accessibility in spatial morphology. The conclusion of this study not only generated great historical value for combing the history of contemporary industrial urban planning in China but is also a significant reference for the sustainable development of the industrial cities in Northeast China.

**Keywords:** planning pattern; industrial base; spatial syntax; connectivity; intelligibility

## 1. Introduction

The spatial morphology of industrial cities in Northeast China was deeply influenced by the Soviet Union's industrial base planning theory and technology in the 1950s. From 1953 to 1957, under the industrial aid of the Soviet Union, a large number of giant industrial bases attached to the main railways were built in the suburban areas of the original cities [1]. With the completion and running of these industrial bases, two positive changes emerged. On one hand, the spatial structure of the original cities expanded and continued; On the other hand, several independent satellite cities have formed with their perfect functional configuration. After 70 years of development, the service functions of these industrial bases are constantly enriched and improved, along with constantly transforming and upgrading their production content and method. Nevertheless, due to the solidification and shackles of the original spatial structure and road network, the sustainable development of these industrial bases is facing great challenges [2].

There were three main benefits to carrying out this study. First of all, the problems of industrial base planning were deeply dissected with the scientific method. Meanwhile, the internal logical relationship between road differentiation and spatial structure was revealed [3]. Secondly, through quantitative study, the spatial function modules with high selectivity and low extension depth were clarified. Based on the above data, we proposed

to modify and improve the spatial distribution of the functional modules in the industrial base to optimize the flow direction of the road network and to eventually meet the needs of industrial production adjustment and future industrial regeneration development [4,5]. Finally, this study can provide more early simulation analysis methods and strategies for new industrial base planning in the future, which makes it more organically and harmoniously coexist with the original city while maintaining sustainable development.

## 2. Literature Review

### 2.1. The Formation Process of the Soviet Union's Industrial Base Planning Pattern and Its Influence on China

During World War II, thousands of industrial buildings in the Soviet Union were damaged. After the war, the "Fourth Five-Year Plan" was implemented in 1946, instigating a large-scale period of national economic recovery. At the same time, the planning pattern paid more attention to economy and rationality, emphasizing the two levels of "overall planning and detailed planning". Large industrial bases planned based on this pattern appeared one after another. A large number of plants were gathered together for their overall planning and design, such as the Novo Krama Polsky Heavy Machine Manufacturing Plant and the Volgograd Tractor Plant [6]. The industrial base planning pattern mainly focused on the scale and layout of industrial land and its relationship with urban land. By arranging industrial chain enterprises closely, the efficient circulation of products and technologies can be realized, and living support service facilities can be combined to improve land-use efficiency. The organization of more complex buildings and functional areas overlapped with each other, which greatly improved the urban planning and design level [7].

Determining how to coordinate the arrangement of industrial land and the overall urban planning was also the key assignment in the planning of important industrial projects in China in the 1950s. The urban planning at that time and the overall urban land planning in the future were centered on urban core industrial projects, and the remaining urban functional land became a supporting supplement around these industrial projects [8]. Therefore, the core industrial projects in this period were not only a new fulcrum in the expansion of an urban spatial structure but also became the main client of urban original functions and infrastructure services. The transfer of industrial technology from the Soviet Union to China at that time led the Soviet Union's industrial base planning theory and technology to have an extremely far-reaching impact on China's urban and rural planning.

### 2.2. The Content of the Soviet Union's Industrial Base Planning Pattern

A series of particular requirements, from principles to technical engineering details, were involved in the Soviet Union's industrial base planning pattern; these were specifically reflected in the following five contents: (1) The land of the industrial base must be compact and reasonable while determining its scale. Excessive land use and scattered workshops will lead to the lengthening of traffic lines and various pipelines in the plant, which will greatly increase construction and operation costs. (2) The distribution of new industrial bases in various economic regions or some cities shall be formulated by the national economic plan. (3) The functional area of the whole industrial base (including the production zone and the living zone) with a large number of buildings in the city should not be arranged on the ground above petroleum minerals and coal mines, as it makes them difficult to mine and brings the possibility of ground deformation; thus, the buildings will be damaged. (4) While delineating the land of an industrial base, it is important to select the transportation method connecting the source of raw materials, the fuels, and the sales market of industrial products. Compared to railway and water transportation, road transportation has more advantages and benefits. For some special industrial plants with wastewater drainage, electric power requirements, and a high-pressure steam supply, relevant engineering facilities of the base shall be considered a priority in general planning to shorten the length of the transmission pipeline. (5) Considering the topographic conditions, it is best to realize the minimum amount of earthwork and foundation works of the construction. The

same requirement should be conducted in the internal traffic facilities and the earthwork of discharge of surface water. All of the above is to ensure a relative balance between excavation and filling [9].

In this period, the Soviet Union launched several specific principles for the planning method aiming at the production zone and living zone in the industrial base. In June 1933, the Soviet Union's Central Executive Committee held a conference about the preparation and approval of the planning and socialist transformation plan for cities and other residential regions of the Soviet Union. The committee members all agreed that "the selection of construction land for a new industrial base or the expansion of existing industrial plant needs to be carried out with the selection of urban service land at the same time" in the resolution. Four specific principles were proposed as follows: (1) The locations of the production zone and the living zone should be considered and arranged to maintain the possibility of cooperation between municipal facilities and water and heat supply systems of the plant. (2) The production zone should not "cut off" or "isolate" the living zone to prevent difficulties in connecting transportation and cultural life services. The production zone should not hinder the development of the living zone. The living zone should avoid disorderly development to make it easier to expand the plant in the future. The new plants should not "fill" the areas along the river to prevent difficulties in using the river as a resource for the residents. (3) While arranging the new plant, it should be considered if there is enough land suitable for building a living zone nearby (outside the protective belt). If the industrial base is arranged near the old origin city, it will hinder its future development. Alternatively, the industrial base may be arranged in a distant rural area that can be built into a nova satellite city serving the old origin city. (4) The sanitary requirements of the production zone and living zone in the industrial base are to ensure that residents avoid the influence of harmful gas, soot, smoke residue, odor, dust, wastewater, and production sounds. The Hygienic Standards and Specifications for the Construction of Industrial Bases (T0CT1324-43) issued by the Soviet Union divided the width of protective belts for hazardous industries into five levels: 1000 m, 500 m, 300 m, 100 m, and 50 m [10,11].

The Soviet Union's industrial base planning pattern strongly emphasizes the expression of ideology. The axis-symmetric planning pattern is the main plane form of production zone and living zone, which reflects the typical aesthetic tendency of socialist realism at that time. The planner divided the production zone and living zone through the main road axis of the industrial base and realized appropriate building scale proportions on both sides of the road axis, which fully displayed the new style of "Socialist City". Its ideological connotation is "carrying out socialist industrial construction, forming a socialist lifestyle, and presenting the grandeur of Socialist City" [12].

### 2.3. The Transfer of the Industrial Base Planning Theory and Technology from the Soviet Union to China

The 1950s was the first "Golden Age" of urban planning in China. China's industrialization was successfully implemented with large-scale technical assistance from the Soviet Union. Several cities in Northeast China began to implement urban planning under this background, which indirectly contributed to the establishment of urban and rural planning theory and construction practice in China. Accompanied by the implementation of a large number of new industrial base projects, the Soviet Union's planning pattern, with its ideology and methodology, had comprehensively entered the planning field of China [13]. In this process, the Soviet Union completely transferred its concepts, content, and standards of the industrial base planning pattern to China through three channels—technical drawings, project guidance, and professional training [14].

Comparing the planning cases of different types of industrial bases in Northeast China, we find that they are permeated with the clear external forms and internal logic of the Soviet Union's pattern. Four specific features were displayed as follows: Firstly, the location of an important core plant determined the direction of urban development and the fulcrum of future spatial expansion. Secondly, based on protecting the old urban area

and historical buildings, the industrial base was connected with the original urban core. The new buildings in the industrial base were designed to show the continuation of the urban style through traffic networks and architectural forms. Thirdly, other land uses were supported around industrial construction in terms of index calculation and site selection. Except for the areas outside the production zone and the old city core, land with good conditions that was close to the industrial zone was selected for the supporting setting of the workers' living zone; Finally, to ensure that the basic living, cultural, health, scientific, and educational needs of workers were met, an emerging industrial city with complete functions and independent operation was established [15].

*2.4. The Application of Quantitative Research in the Development and Regeneration of Industrial Bases*

In the mid-1990s, with the adjustment of industrial production structure, more and more traditional industrial bases gradually lost their production functions and became industrial heritage. Their development and regeneration have been a serious focus in the field of urban planning. Since 2000, several novel mathematical models and statistical technologies have played an increasingly important role in the process of industrial base regeneration planning, which symbolizes that the research on their development and regeneration began to shift from qualitative study to quantitative study [16,17]. The extension depth and connectivity of functional modules and street blocks of the industrial bases were objectively evaluated by applying spatial syntax theory and technology. The business plans and development strategies of cultural and creative industries were formulated based on the reorganization of the architectural spatial modules [18,19]. The internal logical relationship between the road differentiation and the realization of block differentiation was better understood by deeply dissecting the relationship between the syntax of street networks and the differentiation of the size of the blocks in the industrial base. The historical evolution process and future development regulation of the industrial bases will be easily revealed [20]. According to the above research results, the novel extensions of the combined compromised solution methodology, including the logarithmic method, were proposed [21]. In addition, a novel multi-criteria decision-making methodology based on the fuzzy full consistency method and neutrophilic fuzzy measurement alternatives and ranking were developed to establish the compromise solution framework to improve the connectivity and convenience of road network in the industrial bases, which made the realization of the sustainable development of the industrial bases possible [22].

*2.5. Summary*

The emergence, development, and maturity of the Soviet Union's industrial base planning pattern encompass its specific historical background, in which economy, practicality, ideology, and other factors permeate and couple with each other. The advantages of the planning pattern present that the land resources of production chain enterprises were efficiently integrated, a novel pattern of industrial clusters was formatted as the core, railway transportation functions were fully mobilized, and the health environment and functions supporting living zone were improved. These advantages aimed to create a novel socialist life pattern [23,24]. However, the disadvantages are very obvious, as follows: too much attention was paid to the central axis planning pattern to highlight ideology, the rationality of road network was ignored, and subjective consciousness was mixed too much in the planning process, resulting in weak support of the functional modules in the living zone. As a supporting part of the production zone, the location of the living zone did not take into account the impact of the wind environment, sunshine, and noise on the living quality for people, resulting in the deterioration of sustainable development. Over the years, the relevant studies have mainly focused on historical data research, value assessment research, and theoretical and technical research on conservation and regeneration. There is an extreme lack of quantitative research on the spatial morphology characteristics of the industrial base, which leads to the possibility of subjective judgment imbalance in its future

regeneration strategies. The summary of relevant literature research in recent years is listed in Table 1.

**Table 1.** The summary of relevant literature research in recent years.

| Topic of Relevant Research | Main Viewpoints | Scholars and Publication Time |
|---|---|---|
| The planning pattern of the Soviet Union's industrial base | The historical background, the main contents and corresponding standards of the Soviet Union's industrial base planning pattern. | Sun, R. (2013) [5], SCCSU (1975) [6], Niu, X. (2014) [7], Binko, M. (1955) [12], Li, Y. (2010) [16], Tan, Y. (1995) [17], Jevremovic, L. (2014) [24] |
| The transfer of the industrial base planning theory and technology from the Soviet Union to China | The localization process of American industrial technical assistance in the Soviet Union. The channel, content, and process of the transfer of the industrial base planning theory and technology from the Soviet Union to China. The inheritance and innovation of the planning theory and technology in China. | Han, R. (2020) [8], Zhang, Y. (2018) [9], Zhang, B. (2005) [11], Shen, Z. (2002) [14], Wei, L. (2018) [23], |
| Multi-value assessment and application of industrial bases in Northeast China | The qualitative research on historical value, cultural value, aesthetic value, scientific and technological value, and economic value and case application by establishing the multi-value assessment system. | Wu, Y. (2018) [10], Dong, Z. (2004) [13], Zhang, P. (2008) [15] |
| Regeneration and development of the industrial bases | Development strategy and feasibility evaluation method of the regeneration of industrial heritage. Research on the theory and technology to serve the regeneration. | Han, R. (2020) [1], Li, L. (2021) [3], Yang, J. (2018) [4], Guo, F. (2015) [18], Liu, X. (2019) [19] |
| Development planning of the industrial bases based on quantitative research | The internal logical relationship between the road differentiation and the block differentiation. Algorithms for improving the connectivity of the road network. | Liu, F. (2019) [2], Lim, L. (2015) [20], Deveci, W. (2010) [21], Pamucar, D. (2021) [22] |

Since the 1950s, while absorbing the advanced planning technology of the Soviet Union, the industrial bases in Northeast China inevitably inherited many drawbacks, which led to lots of problems for the sustainable development of urban spatial morphology over the next 70 years. In order to improve the sustainability of these industrial bases, in this study, we introduced spatial syntax technology to conduct a quantitative analysis of functional modules and road axes, thus revealing the origin of the defects of the Soviet Union's industrial base planning pattern, establishing optimization strategies, and artificially improving the sustainability of the industrial bases by reorganizing road flowing patterns and changing the land uses of functional modules.

## 3. Methods

### 3.1. Background and Cases Selection

In the process of planning practice, industrial base planning not only implements the planner's subjective design intention but also follows an objective internal order law. This order law can be interpreted and described through a scientific calculation method while a certain language for the spatial order law is found; then, space is organized into a spatial

sequence that is easily understood and recognized. The theoretical basis of spatial syntax is established upon the analysis of spatial visual connection in the previous urban planning theory and a deep study on the factors of people's perception of space. Although the guidance of space can shape human behavior, the uncertainty of human activities requires scientific data to study their commonalities and benchmark values. The data-based analysis of spatial syntax produces more benchmark values, which provide a basis for research on the interaction between workers and industrial base space [25].

Spatial syntax was first proposed by Bill Hillier of University College London in the 1970s. It described the spatial morphological characteristics and mutual relationships of roads, buildings, and landscapes through quantitative analysis. According to different research objects and their spatial forms, this study mainly applied two research methods of spatial syntax—convex map and axial map. They included six main indexes—choice, total depth, connectivity, integration, synergy, and intelligibility [26].

The history of urban construction in Northeast China is closely related to the planning of industrial bases in the 1950s, which had a far-reaching influence on the evolution of urban spatial structure. There are three important industrial cities in Northeast China. This study selected the following three representative cases in the above three cities: (1) the Harbin Three Power Industrial Base (HTPIB), representing the new industrial base of a multi-plant joint type, which consists of the Harbin Electric Machinery Plant (HEMP), the Harbin Boiler Plant (HBP), and the Harbin Turbine Plant (HTP) [27]; (2) the Changchun First Automobile Works Base (CFAWB), representing the new industrial base of a single plant, Changchun First Automobile Works; (3) the Shenyang Dadong Aeronautics and Astronautics Industrial Base (SDAAIB), representing the reconstructed and expansion industrial base of embedding, consisting of the 111 Plant and the 410 Plant [28]. Based on the collection of GIS data and the optimization of different functional module forms and road axis networks in these three industrial bases, this study made the cases comply with the calculation rules of spatial syntax to conduct the simulation analysis. The general planning drawings of the above three industrial bases are shown in Figures 1–3, respectively.

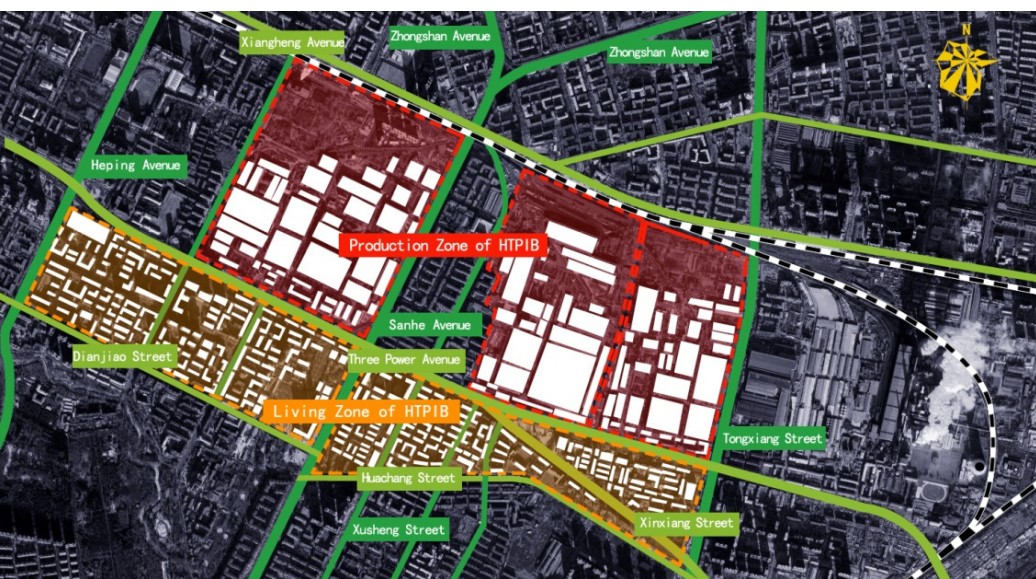

**Figure 1.** The general planning of Harbin Three Power Industrial Base.

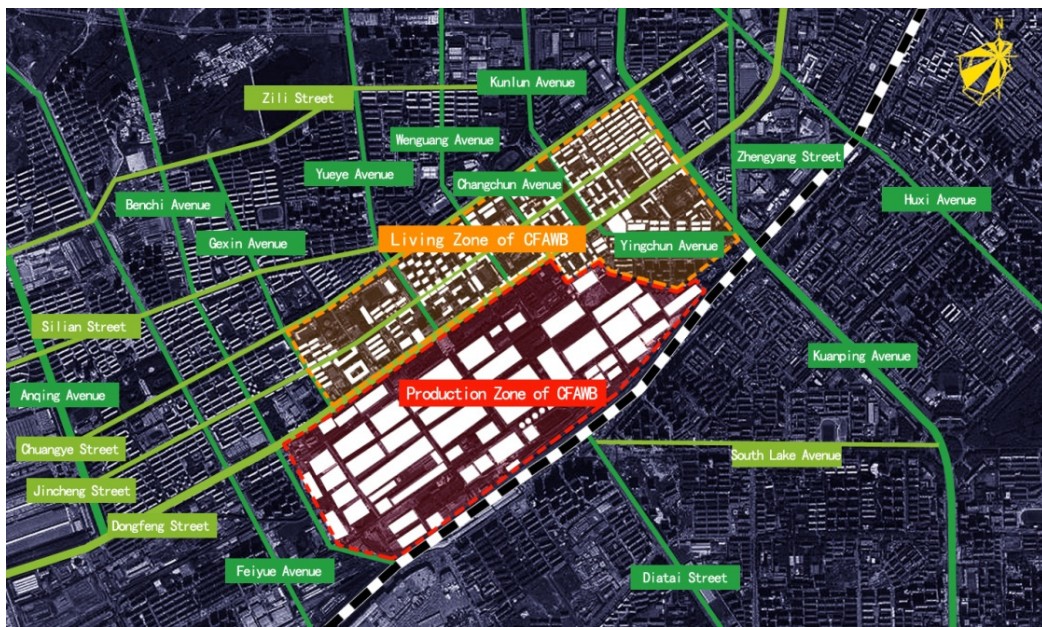

**Figure 2.** The general planning of Changchun First Automobile Works Base.

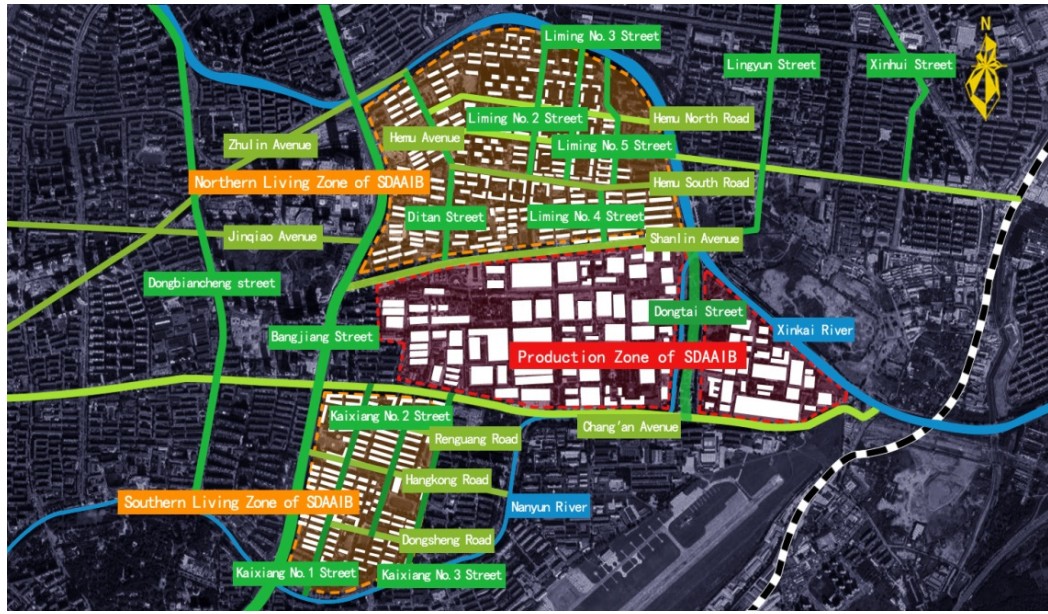

**Figure 3.** The general planning of Shenyang Dadong Aeronautics and Astronautics Industrial Base.

*3.2. Simulation*

3.2.1. Simulation of Spatial Extension and General Planning Analysis

Based on the general planning drawings of the three industrial bases, the spatial structure of each functional module of the production zone and living zone was described using an optimized two-dimensional graphical language. This study used convex mapping to conduct the simulation analysis and obtained statistical values with Depthmap Beta® 1.0 (Shenzhen University, team ARI, Shenzhen, China). The spatial extension simulation and linkage of the three industrial bases are shown in Figures 4–6, respectively.



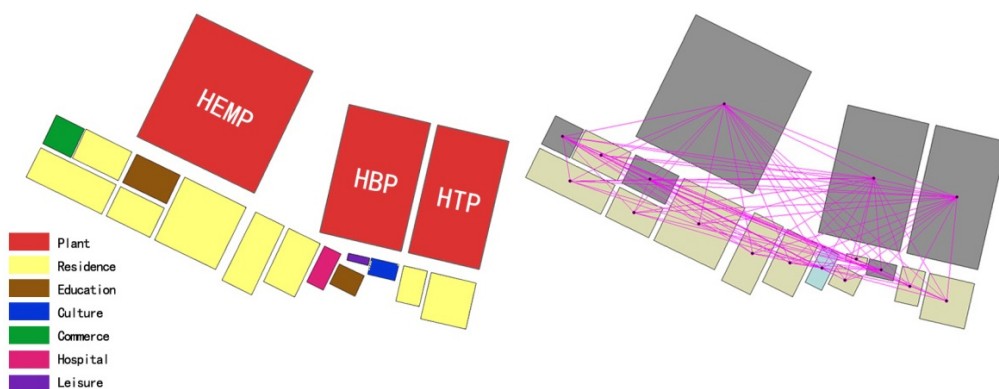

**Figure 4.** The spatial extension simulation of Harbin Three Power Industrial Base.

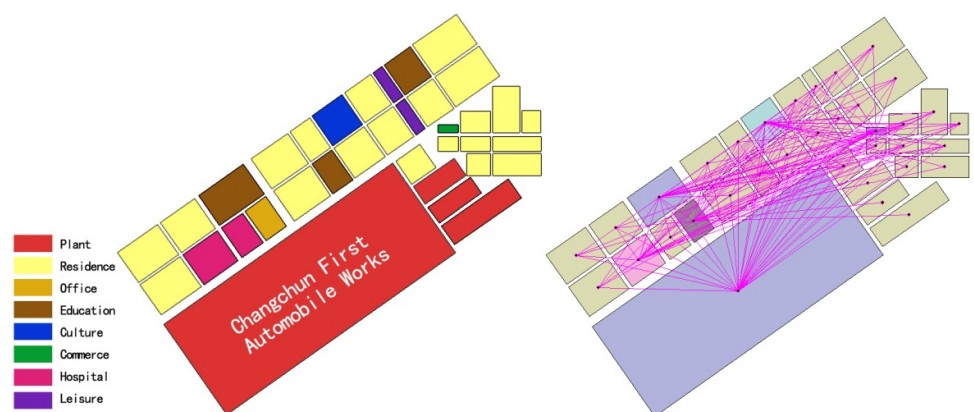

**Figure 5.** The spatial extension simulation of Changchun First Automobile Works Base.

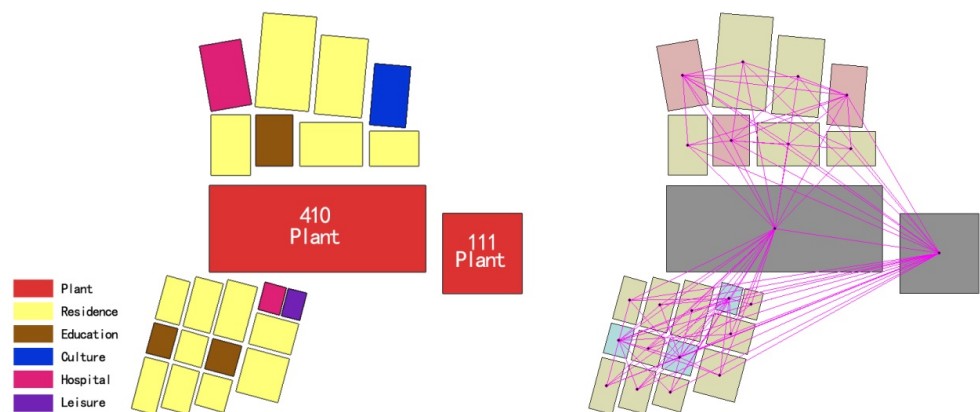

**Figure 6.** The spatial extension simulation of Shenyang Dadong Aeronautics and Astronautics Industrial Base.

Through the optimization of spatial structure graphics and the linkage simulation of functional modules, it is observed that the production zone of the Harbin Three Power Industrial Base is composed of three independent plants. The living zone is located in the south of the production zone. The residence module, hospital module, education module, culture module, and other modules are distributed linearly, and the connectivity among each module is general. The Changchun First Automobile Works Base is composed of a single plant. The living zone is located in the north of the production zone. The residence module, hospital module, education module, culture module, and other modules are distributed in the form of rows and columns, and the connectivity among various functional modules is strong. The Shenyang Dadong Aeronautics and Astronautics Industrial Base is

composed of two plants. The living zones are distributed on the north and south sides of the production zone. The residence module, hospital module, education module, culture module, and other modules are interspersed on both sides of the living zones, and the connectivity among each module is weak. Under the space extension simulation, the production zone of the Changchun First Automobile Works Base is in a good position, forming the auxiliary function distribution around it, which has the best support effect for the behavior of workers from the living zone to the production zone.

### 3.2.2. Simulation of Road Accessibility and Road Planning Analysis

The road network is an important supporting element of spatial syntax theory. From another perspective, it reflects the spatial circulation and accessibility outside the space occupied by functional modules under a specific planning pattern. The three cases have similar commonalities in the planning of road networks as well as their own unique features. The optimized road connection in the industrial base was described using a two-dimensional graphical language. This study used axial mapping to conduct simulation analysis and obtained statistical values with Depthmap Beta® 1.0 (Shenzhen University, team ARI, Shenzhen, China). The road network accessibility simulations of the three industrial bases are shown in Figures 7–9, respectively.

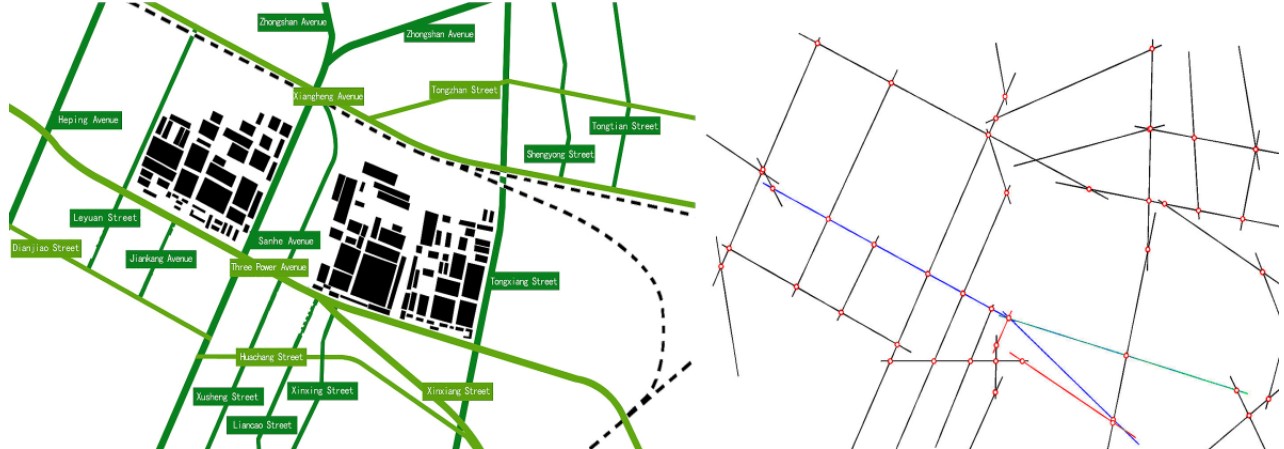

**Figure 7.** The road network accessibility of Harbin Three Power Industrial Base.

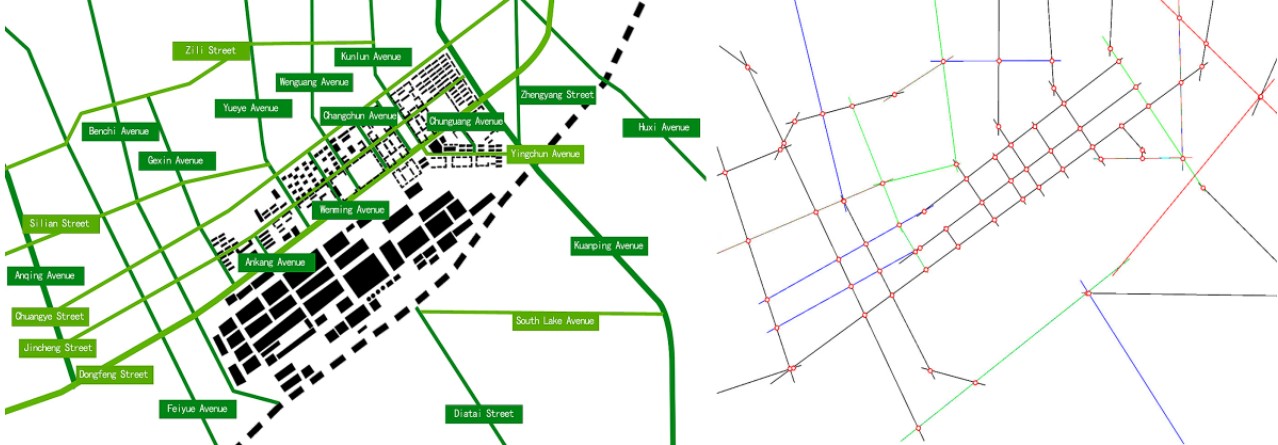

**Figure 8.** The road network accessibility of Changchun First Automobile Works Base.

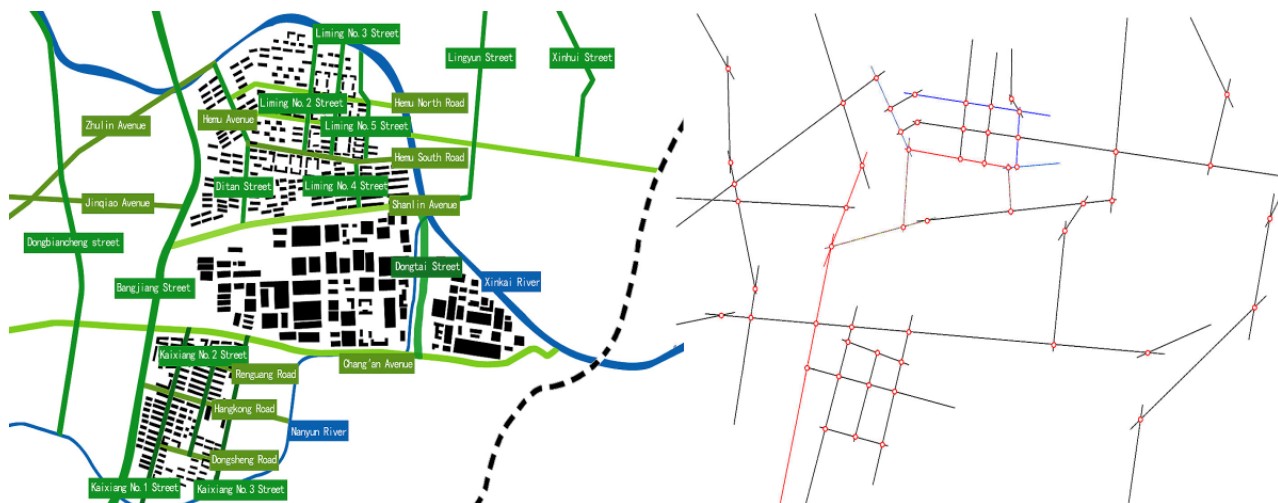

**Figure 9.** The road network accessibility of Shenyang Dadong Aeronautics and Astronautics Industrial Base.

Through the optimization of road axis graphics, it was observed that the road network structure of the Harbin Three Power Industrial Base is the simplest. The production zone and living zone are divided by Three Power Avenue, and the internal roads in the living zone are connected to Three Power Avenue through secondary urban roads, showing that the living zone has efficient support and connectivity to the production zone. The road network structure of the Changchun First Automobile Works Base is complex and orderly. The production zone and living zone are divided by Dongfeng Street, the main urban road. Jincheng Street and Chuangye Street further divide the living zone, forming a traffic network inside the living zone. This planning pattern not only realizes effective support for the production zone but also achieves good connectivity of the internal functional modules of the living zone. The road network structure of the Shenyang Dadong Aeronautics and Astronautics Industrial Base is complex and disordered. The north living zone, production zone, and south living zone are divided by Shanlin Avenue and Chang'an avenue, in which Chang'an Avenue connects downtown Shenyang City and the whole industrial base. The road network in both the north and south living zones is quite disorderly, which not only weakens support for the production zone but also fails to achieve the connectivity of the main functional modules distributed in both living zones.

## 4. Data Analysis and Discussion

### 4.1. Analysis for the Functional Realization of the Spatial Module Based on Convex Mapping

The main data analysis of convex mapping selects three indexes—choice, total depth, and connectivity, presenting the shortest topological path times, maximum topological depth, and achievable connection selection number of a single functional module in the whole industrial base, respectively. The statistical results generated by simulation calculation can further reveal the realization level of functional modules in different spatial locations under specific planning patterns.

Choice indicates the possibility of a spatial node being selected or the number of times that a node appears on the shortest topological path. This index can be used to measure the potential of crossing traffic attracted by a spatial element. Total depth represents the accessibility of spatial nodes in the topological sense—that is, the portability of nodes in the spatial system. Connectivity refers to the number of nodes connected with a spatial node. The higher the connectivity, the higher the possibility of spatial connection to the outside [29]. The functional realization index statistics of the three industrial bases are listed in Tables 2–4, respectively.

**Table 2.** The functional realization index statistics of Harbin Three Power Industrial Base.

| Name of Module | Choice | Total Depth | Connectivity |
|---|---|---|---|
| HEMP Module | 10 | 0 | 16 |
| HBP Module | 3 | 0 | 16 |
| HTP Module | 9 | 0 | 16 |
| Residence Module (Average) | 10 | 25 | 7 |
| Education Module (Average) | 9 | 0 | 16 |
| Culture Module | 9 | 0 | 16 |
| Commerce Module | 28 | 0 | 16 |
| Hospital Module | 24 | 17 | 15 |
| Leisure Module | 10 | 26 | 7 |

**Table 3.** The functional realization index statistics of Changchun First Automobile Works Base.

| Name of Module | Choice | Total Depth | Connectivity |
|---|---|---|---|
| CFAW Module | 74 | 39 | 29 |
| Residence Module (Average) | 10 | 61 | 9 |
| Office Module | 396 | 0 | 34 |
| Education Module (Average) | 48 | 39 | 29 |
| Culture Module | 97 | 38 | 30 |
| Commerce Module | 94 | 39 | 29 |
| Hospital Module | 111 | 41 | 27 |
| Leisure Module | 0 | 61 | 7 |

**Table 4.** The functional realization index statistics of Shenyang Dadong Aeronautics and Astronautics Industrial Base.

| Name of Module | Choice | Total Depth | Connectivity |
|---|---|---|---|
| 111 Plant Module | 131 | 0 | 22 |
| 410 Plant Module | 130 | 0 | 22 |
| Southern Residence Module (Average) | 7 | 39 | 5 |
| Northern Residence Module (Average) | 5 | 39 | 5 |
| Southern Education Module | 9 | 30 | 15 |
| Northern Education Module | 6 | 39 | 10 |
| Southern Hospital Module | 9 | 30 | 15 |
| Northern Hospital Module | 5 | 35 | 10 |
| Culture Module | 12 | 35 | 9 |
| Leisure Module | 4 | 39 | 6 |

Through simulation calculation analysis and data statistics, it can be seen that the production zones of the Harbin Three Power Industrial Base, the Changchun First Automobile Works Base, and the Shenyang Dadong Aeronautics and Astronautics Industrial Base are located in suitable spatial locations; thus, other living functional modules, such as the residence module, education module, and culture module, have good connectivity. Taking the production zones of the three industrial bases as the expansion center, the steps of the other functional modules are at a low level, indicating that they have a good realization for the different needs of the people who work and live there.

In the planning practice of the living zone of the three cases, the residence module in the Changchun First Automobile Works Base has the best accessibility (Choice: 10, Total Depth: 61, Connectivity: 9), and has strong support for the living needs of workers. However, the leisure module (Choice: 0, Total Depth: 61, Connectivity: 7), which provides public leisure and entertainment, is in a relatively unfavorable spatial location and has poor support for the other modules in the living zone. Due to historical reasons, the living zones in the Shenyang Dadong Aeronautics and Astronautics Industrial Base were built in batches in the north and south. Despite having an independent residence module, education module, and hospital module, their spatial organization characteristics lead to the tearing of their interaction. Especially the culture module (Choice: 12, Total Depth: 35, Connectivity: 9) and the leisure module (Choice: 4, Total Depth: 39, Connectivity: 6) have poor common support for the production zone and living zone. According to the above analysis, it is shown that in the process of the transfer of the industrial base planning theory and technology from the Soviet Union to China in the 1950s, the planning pattern of the new industrial base of a single plant is more relatively reasonable and scientific, which achieved good support for both the production zone and living zone.

### 4.2. Comparison of Road Synergy and Intelligibility Based on Axial Mapping

The main data analysis of axial mapping selects three indexes—integration (including global integration and local integration), synergy, and intelligibility, presenting the accessibility of different classes of roads in the region. It further reveals the supporting level of the three industrial bases for the production zone and living zone reflected by the differences in road network organization under different planning patterns.

Integration is an important index to measure spatial permeability, which presents the level of aggregation or dispersion between one space and other spaces in the system. The higher the level of integration, the more aggregated it is between contiguous spaces; the lower the level of integration, the more scattered it is between contiguous spaces. Global integration indicates the tightness of a road with the whole spatial system; local integration indicates the tightness of a road with other roads within a certain topological area. Synergy is a variable describing the correlation between global integration and local integration. A higher synergy level of an urban space indicates that the local center of the urban space can be better integrated into the whole urban space system. Intelligibility is a variable used to measure the correlation between global integration and connectivity [30]. The synergy and intelligibility of different classes of roads for the three industrial bases are listed in Tables 5–7, respectively.

**Table 5.** Synergy and intelligibility of different classes roads for Harbin Three Power Industrial Base.

| Index | Abscissa and Ordinate | Correlation | Correlation Image |
|---|---|---|---|
| Synergy (R = 3) | X = Integration[H][H] Y = Integration[H][H]R3 | $R^2 = 0.918$ Y = 1.287X + 0.124 |  |
| Synergy (R = 5) | X = Integration[H][H] Y = Integration[H][H]R5 | $R^2 = 0.996$ Y = 0.963X + 0.067 |  |
| Intelligibility | X = Connectivity Y = Integration[H][H] | $R^2 = 0.673$ Y = 0.149X + 0.842 |  |

**Table 6.** Synergy and intelligibility of different classes roads for Changchun First Automobile Works Base.

| Index | Abscissa and Ordinate | Correlation | Correlation Image |
|---|---|---|---|
| Synergy (R = 3) | X = Integration[H][H]<br>Y = Integration[H][H]R3 | $R^2$ = 0.968<br>Y = 0.952X + 0.262 |  |
| Synergy (R = 5) | X = Integration[H][H]<br>Y = Integration[H][H]R5 | $R^2$ = 0.999<br>Y = 0.999X + 0.022 |  |
| Intelligibility | X = Connectivity<br>Y = Integration[H][H] | $R^2$ = 0.747<br>Y = 0.170X + 1.082 |  |

**Table 7.** Synergy and intelligibility of different classes roads for Shenyang Dadong Aeronautics and Astronautics Industrial Base.

| Index | Abscissa and Ordinate | Correlation | Correlation Image |
|---|---|---|---|
| Synergy (R = 3) | X = Integration[H][H]<br>Y = Integration[H][H]R3 | $R^2$ = 0.623<br>Y = 1.500X + 0.003 |  |
| Synergy (R = 5) | X = Integration[H][H]<br>Y = Integration[H][H]R5 | $R^2$ = 0.826<br>Y = 0.954X + 0.221 |  |
| Intelligibility | X = Connectivity<br>Y = Integration[H][H] | $R^2$ = 0.330<br>Y = 0.086X + 0.723 |  |

The scatter chart on the right side of Table 5 presents the correlation between the density of the road network and integration. The coupling relationship of the scatter chart conforms to the following equation:

$$Y = aX + b \tag{1}$$

We obtained the data and the straight-line graph with the intelligent simulation of Depthmap Beta® 1.0 (Shenzhen University, team ARI, Shenzhen, China). The average global integration value of the road system in the Harbin Three Power Industrial Base is 1.345, the highest value is 2.064, and the lowest value is 0.800. The highest one is distributed around the Harbin Electric Machinery Plant (HEMP), including Three Power Avenue (Integration: 1.976), Sanhe Avenue (Integration: 2.063), and Xiangheng Avenue (Integration: 1.988), indicating that this area has the strongest accessibility in the overall road network of the industrial base and is the core of the road traffic system in the base.

We conducted a calculation of the synergy level between the global integration (R = n) and the local integration (R = 3, 5, 7, 9, 11). The relationship of synergy level is defined as $R^2$. The higher the $R^2$ value, the higher the correlation between the global integration and the local integration. $R^2 > 0.5$ means that there is a correlation; $R^2 > 0.7$ means that there is an

intense relationship between them. According to the data analysis in Table 4, the synergy values of "Radius-3" (R = 3) and "Radius-5" (R = 5) of the Harbin Three Power Industrial Base are 0.918 and 0.996, respectively, which are highly correlated, indicating that the local centers in different areas are better integrated into the whole industrial base. Through the observation of the correlation image between connectivity and global integration, we found that the intelligibility value $R^2 = 0.673$ (<0.7), indicating that the correlation between connectivity and the global integration level is general; the intelligibility is also general. In addition, it can be seen that the roads with a high integration value express high connectivity and a high synergy level; meanwhile, other roads with a low integration value have low compatibility and a low synergy level.

The average value of global integration of the Changchun First Automobile Works Base is 1.783, the highest value is 3.739, and the lowest one is 1.103, indicating that the integration level of the roads in this industrial base is extremely high, the production zone and living zone are relatively close as a whole, and there is an intense connection between different classes of roads. The main road in the base, Dongfeng Street (Integration: 3.739), has the highest integrated value and connects many secondary roads with good connections. As a result, the functional modules along Dongfeng Street form better support for the production zone and living zone. Jincheng Street (Integration: 2.447) and Chuangye Street (Integration: 2.281), which are arranged parallel to Dongfeng Street, also achieve good compactness, providing better connectivity for the living zone.

According to the data analysis in Table 5, the result $R^2$ is greater than 0.7 under the situation of the synergy values of R = 3 and R = 5, indicating that the planning pattern of the Changchun First Automobile Works Base has a high correlation. Its intelligibility $R^2 = 0.747$ (>0.7) indicates that the correlation between connectivity and global integration is extremely intense. The above data suggest that the main roads and secondary roads in this base have good guidance; meanwhile, the overall spatial planning has good regularity. The advantages of the Soviet Union's planning pattern in the practice of the new industrial base of a single plant are shown to be most prominent in this case.

The average global integration value of the Shenyang Dadong Aeronautics and Astronautics Industrial Base is 0.990, the highest value is 1.401, and the lowest one is 0.495. The roads with high global integration value are mainly distributed in the south living zone, including Chang'an Street (Integration: 1.401), Kaixiang 1st Road (Integration: 1.329), and Hangkong Road (Integration: 1.449), indicating that these roads have the strongest accessibility in the road network of the base.

According to the data analysis in Table 6, the result $R^2$ is less than 0.7 under the synergy value of R = 3 and the result $R^2$ is greater than 0.7 under the situation of the synergy value R = 5, indicating that the correlation of roads in this base is general. The intelligibility of the roads, $R^2 = 0.330$ (<0.5), indicates that the correlation between connectivity and global integration is quite weak. The above data suggest that the main roads and secondary roads in this base have worse guidance; in addition, the overall spatial planning has poor regularity.

## 5. Conclusions

Through a literature review, this study outlined the development process of the Soviet Union's industrial base planning pattern, its content and principles, and its transfer to Northeast China, where the most important industrial bases are distributed. China's urban planning and spatial morphology have been greatly influenced by the Soviet Union's industrial base planning theory and technology developed in the 1950s. Different planning practice patterns with distinctive characteristics and coexistence of advantages and disadvantages have been implemented in three important cases. The Harbin Three Power Industrial Base (HTPIB) represents the new industrial base of a multi-plant joint type; the Changchun First Automobile Works Base (CFAWB) represents the new industrial base of a single plant; the Shenyang Dadong Aeronautics and Astronautics Industrial Base (SDAAIB) represents the reconstructed and expanded industrial base of embedding. The

three cases have integrated planning of production zones and living zones, as well as specific characteristics in functional module distribution and road network structure.

With the simulation of spatial extension and road accessibility along with data analysis, this study comprehensively, rationally, and objectively evaluated the planning practice level of the three cases and revealed the coupling relationship between spatial morphology and functional realization. Our conclusions are as follows:

Through the space extension simulation based on convex mapping, the production zone of the Changchun First Automobile Works Base is in an ideal position, forming the auxiliary function distribution around it, which has the best support effect for the behavior of workers from the living zone to the production zone. We produced statistics and analyzed the obtained data and found that the residence module in the Changchun First Automobile Works Base has the best accessibility and strong support for the living needs of workers, which indicates that the planning pattern of the new industrial base of a single plant is relatively more reasonable and scientific than the other two planning patterns and achieved good support for both the production zone and the living zone.

Through the road connectivity simulation based on axial mapping, it was observed that the road network structure of the Changchun First Automobile Works Base is complex and orderly. Its average value of global integration (1.783) is the highest. The Harbin Three Power Industrial Base has the second highest value (1.345), and the Shenyang Dadong Aeronautics and Astronautics Industrial Base has the lowest value (0.990). The above data present that the integration level of the roads in the Changchun First Automobile Works Base is extremely high, the production zone and living zone are relatively close as a whole, and there is an intense connection between different classes of roads. The advantages of the Soviet Union's planning pattern in the practice of the new industrial base of a single plant are the most prominent in this case.

Although spatial syntax technology provides a visible and objective analysis method, there are two limitations in the process of the simulation of spatial extension and road accessibility. Firstly, both convex mapping and axial mapping can only describe the two-dimensional relationship of the spatial morphology and ignore the consideration of the factors in three-dimensional space, such as the height of buildings, ground elevation changes, and attraction of the color and noise of the space. Secondly, mandatory functions of block modules (schools, businesses, and hospitals) and regulations and the management of the traffic can influence people's behavior and vehicle flow along with getting rid of the passive influence of spatial morphology. The two limitations result in the reduction of simulation analysis accuracy. In addition, in the 1950s, other regions of China also conducted large-scale industrial construction under the guidance of the Soviet Union's industrial planning patterns, such as the southeast and the southwest. In order to better understand the practice level, we will continue our research to select more representative cases from other regions to implement a comparative study.

For the sustainable development of the industrial cities in Northeast China, we recommend the following points:

- To improve the sustainability of the new emerging industrial cities, greater adoption of the planning pattern for the new industrial base of a single plant more than the new industrial base of a multi-plant joint type is encouraged, as it is easier to form a reasonably independent, circular, and small satellite city, and it will not affect the expansion and development of urban spatial structures in the future. At the beginning of the planning, we suggest determining the number and size of block modules and dividing the living zone according to the functional requirements with different level. Firstly, the simulation of spatial extension and road accessibility along with data analysis should be conducted for several comparative planning plans; the land uses of block modules for different needs will be determined. Secondly, the road connectivity could be simulated based on the street grid established by the block module distribution. According to the above analysis results, the classes and widths of

the roads could be determined. The traffic rules can even be determined to restrict the direction of traffic flow and better support the mandatory functions of block modules.

- Based on the simulation and analysis of spatial syntax, we could moderately excavate and develop the industrial tourism resources of the industrial base cases. On one hand, we could add supporting services to the functional modules with the best accessibility (high choice value, low total depth value) in the living zone; on the other hand, we could change the usage properties of the buildings and land sites with worse accessibility (high total depth value, low connectivity value). In doing so, the functional modules in the living zone can be efficiently integrated and established as service facilities around the theme of industrial tourism. According to the data analysis results of road connectivity, the areas with high integration are defined, where the surrounding roads have better connectivity to the whole industrial base. Traffic flow and people flow should be investigated to conform to the main functional spaces of industrial tourism, such as the industrial culture exhibition space, industrial technology exhibition space, and conference business space. Considering the respective unique industrial characteristics, we suggest that the three cases in Northeast China can be purpose developed for different industrial tourism projects with specific themes.

**Author Contributions:** Conceptualization: R.H.; investigation: R.H and D.L.; methodology: G.Z. and D.L.; writing—original draft: R.H.; review and editing: R.H. and L.L. All authors have read and agreed to the published version of the manuscript.

**Funding:** This research was funded by the Key Science and Technology Project of the Jilin Provincial Department of Science and Technology (20210203142SF), the Key Scientific Research Project of the Jilin Provincial Department of Education (JJKH20210277KJ).

**Informed Consent Statement:** Informed consent was obtained from all subjects.

**Acknowledgments:** We would like to thank the anonymous reviewers for their constructive and supportive feedback.

**Conflicts of Interest:** The authors declare no conflict of interest.

## Abbreviations

| | |
|---|---|
| HTPIB | Harbin Three Power Industrial Base |
| CFAWB | Changchun First Automobile Works Base |
| SDAAIB | Shenyang Dadong Aeronautics and Astronautics Industrial Base |
| HEMP | Harbin Electric Machinery Plant |
| HBP | Harbin Boiler Plant |
| HTP | Harbin Turbine Plant |

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
