# Peer review of "A Comparative Study on Planning Patterns of Industrial Bases in Northeast China Based on Spatial Syntax"

_sustainability, doi:10.3390/su14021041_

Round 1

Reviewer 1 Report

This paper focuses the development process of the industrial base planning pattern created by the Soviet Union in the 1950s. This study is interesting. My comments are as below.

  1. The introduction is not very useful, therefore, the introduction should be extended very carefully, so, the introduction section should be rewritten again, the introduction should highlight the novelty and motivation of study, not only put some literature without any useful explanation, in fact, the introduction should be clearly stated research questions and targets first. Then answer several questions: Why is the topic important (or why do you study on it)? What are the research questions? What has been studied? What are your contributions? Why is to propose this particular method?
  2. At the end of literature review you should come out with a paragraph to conclude your discussion, in this paragraph you can highlight the novelty of your study.
  3. I advise authors to present the current literature and their contribution to the literature with a summary table.
  4. I advise authors to clearly explain why they have preferred to improve Spatial syntax method. What was their motivation to implement Spatial syntax and what benefits they have seen comparing the other methods?
  5. The literature review can be enriched with the following studies: (i) Assessment of alternative fuel vehicles for sustainable road transportation of United States using integrated fuzzy FUCOM and neutrosophic fuzzy MARCOS methodology. Science of The Total Environment, 788, 147763. (ii) Fuzzy Power Heronian function based CoCoSo method for the advantage prioritization of autonomous vehicles in real-time traffic management. Sustainable Cities and Society, 69, 102846.
  6. The authors need to discuss about the limitations of the proposed method as well as case study limitations, what are your recommendations for future works, how the proposed method solved the case study problem.
  7. How practitioners can use the proposed method in the real life problems, how the proposed method is useful for future studies.

Author Response

Dear reviewer,

Thank you for handling the review of our manuscript entitled “A Comparative Study on Planning Patterns of Industrial Bases in Northeast China based on Spatial Syntax (Sustainability-1518321)”. In this revised version, we have carefully addressed all the issues raised by you and invited a native speaker to proofread the whole manuscript before resubmission (The English-Editing-Certification is in the attachment of this cover letter). We sincerely appreciate your insightful comments and suggestions that greatly help improve our manuscript.

The following is a summary of the point-to-point response and revisions we have made to each of your comments. We look forward to hearing your more suggestion and the outcome of this latest version of the manuscript.

Point 1: The introduction is not very useful, therefore, the introduction should be extended very carefully, so, the introduction section should be rewritten again, the introduction should highlight the novelty and motivation of study, not only put some literature without any useful explanation, in fact, the introduction should be clearly stated research questions and targets first. Then answer several questions: Why is the topic important (or why do you study on it)? What are the research questions? What has been studied? What are your contributions? Why is to propose this particular method?

Response 1: Thank you for your suggestion which benefits us a lot. We have rewritten the introduction.

(Line 65-77) " The spatial morphology of industrial cities in Northeast China has been deeply influenced by the Soviet Union's industrial base planning theory and technology in the 1950s. From 1953 to 1957, under the industrial aid of the Soviet Union, a large number of giant industrial bases attaching to the main railways were built in the suburban areas of the original cities [1]. With the completion and running of these industrial bases, two positive changes emerged. On one hand, the spatial structure of the original cities has been expanded and continued; On the other hand, several independent satellite cities have been formed with their perfect functional configuration. After 70 years of development, the service functions of these industrial bases are constantly enriched and improved, along with constantly transforming and upgrading their production content and method. Nevertheless, due to the solidification and shackles of the original spatial structure and road network, the sustainable development of these industrial bases is facing great challenges [2, 3]."

(Line 78-89) " There are three main benefits to carrying out this study. First of all, the problem of the industrial base planning was deeply dissected with the scientific method, meanwhile, the internal logical relationship between road differentiation and spatial structure was revealed [4]; Secondly, Through quantitative study, the spatial function modules with high selectivity and low extension depth were clarified. Based on the above data, we proposed to modify and improve the spatial distribution of the functional modules in the industrial base, to optimize the flow direction of the road network, to eventually meet the needs of industrial production adjustment and future industrial regeneration development [5, 6]; Finally, this study can provide more early simulation analysis methods and strategies for the new industrial base planning in the future, which makes it more organically and harmoniously coexist with the original city while its sustainable development."

Point 2: At the end of literature review you should come out with a paragraph to conclude your discussion, in this paragraph you can highlight the novelty of your study.

Response 2: We totally agree with your view. We came out a new section "2.5. Summary" to conclude our discussion .

(Line 227-246) "The emergence, development, and maturity of the Soviet Union's industrial base planning pattern have its specific historical background, in which economy, practicality, ideology, and other factors permeate and couple with each other. The advantages of the planning pattern present that the land resources of production chain enterprises were efficiently integrated, a novel pattern of industrial clusters was formatted as the core, railway transportation functions were fully mobilized, the health environment and functions supporting living zone was improved. These advantages aim to create a novel socialist life pattern [24, 25]. However, the disadvantages are very obvious as follows: paying too much attention to the central axis planning pattern to highlight ideology, ignoring the rationality of road network, mixing too much subjective consciousness in the planning process, and resulting in weak support of the functional modules in the living zone. As a supporting part of the production zone, the location of the living zone did not take into account the impact of the wind environment, sunshine, and noise on the living quality for people, resulting in the deterioration of sustainable development. Over the years, the relevant studies have mainly focused on historical data research, value assessment research, theoretical and technical research on conservation and regeneration. There exists an extreme lack of quantitative research on the spatial morphology characteristics of the industrial base, which leads to the possibility of subjective judgment imbalance in its future regeneration strategies. The summary of relevant literature research in recent years is listed in Table 1."

Additionally, We have added a paragraph to highlight the novelty of our study at the end of this section

(Line 248-256) " Since the 1950s, while absorbing the advanced planning technology of the Soviet Union, the industrial bases in Northeast China inevitably inherited many drawbacks, which leads to lots of problems for sustainable development of urban spatial morphology in the next 70 years. In order to improve the sustainability of these industrial bases, this study introduces spatial syntax technology to conduct quantitative analysis of functional modules and road axes, thus revealing the origin of the defects of the Soviet Union's industrial base planning pattern, establishing optimization strategies, and artificially improving the sustainability of the industrial bases by reorganizing roads flowing pattern and changing the land uses of functional modules."

Point 3: I advise authors to present the current literature and their contribution to the literature with a summary table.

Response 3: We came out a new Table 1. to present the current literature and their contribution.

(Line 247-248)

Table 1. The summary of relevant literature research in recent years.

Topic of Relevant Research

Main Viewpoints

Scholars and Publication Time

The planning pattern of the Soviet Union's industrial base

The historical background, the main contents and corresponding standards of the Soviet Union's industrial base planning pattern.

Binko, M.(1955), SCCSU(1975), Tan, Y.(1995), Li, Y.(2010), Sun, R.(2013), Niu, X.(2014), Jevremovic, L.(2014)

The transfer of the industrial base planning theory and technology from the Soviet Union to China

The localization process of American industrial technical assistance in the Soviet Union. The channel, content, and process of the transfer of the industrial base planning theory and technology from the Soviet Union to China. The inheritance and innovation of the planning theory and technology in China.

Shen, Z.(2002), Zhang, B.(2005), Zhang, Y.(2018), Wei, L.(2018),  Han, R.(2020),

Multi-value assessment and application of industrial bases in Northeast China

The qualitative research on historical value, cultural value, aesthetic value, scientific and technological value, and economic value. case application by establishing the Multi-value assessment system.

Dong, Z.(2004), Zhang, P.(2008), Wu, Y.(2018)

Regeneration and development of the   industrial bases

Development strategy and feasibility evaluation method of the regeneration of industrial heritage. Research on the theory and technology to serve the regeneration.

Guo, F.(2015), Yang, J.(2018), Han, R.(2020), Liu, X.(2019),   Li, L.(2021),

Development planning of the industrial bases based on quantitative research

The internal logical relationship between the road differentiation and the block differentiation. Algorithms for improving the connectivity of the road network.

Deveci,W.(2010), Lim, L.(2015), Liu, F.(2019), Pamucar,D.(2021)

Point 4: I advise authors to clearly explain why they have preferred to improve Spatial syntax method. What was their motivation to implement Spatial syntax and what benefits they have seen comparing the other methods?

Response 4: We have added a new paragraph to discuss the motivation to implement spatial syntax and its benefits.

(Line 203-218) "In the mid-1990s, with the adjustment of industrial production structure, more and more traditional industrial bases gradually lost their production functions and became industrial heritage. Their development and regeneration have been seriously focused in the field of urban planning. Since 2000, several novel mathematical models and statistical technology have played an increasingly important role in the process of industrial base regeneration planning, which symbolized that the research on their development and regeneration began to shift from qualitative study to quantitative study [17, 18]. The extension depth and connectivity of functional modules and street blocks of the industrial bases were objectively evaluated by applying spatial syntax theory and technology. The business plans and development strategies of cultural and creative industries were formulated based on the reorganization of the architectural spatial modules [19, 20]. The internal logical relationship between the road differentiation and the realization of block differentiation was better understood by deeply dissecting the relationship between the syntax of street networks and the differentiation of the size of the blocks in the industrial base. The historical evolution process and future development regulation of the industrial bases will be easily revealed [21].

Point 5: The literature review can be enriched with the following studies: (i) Assessment of alternative fuel vehicles for sustainable road transportation of United States using integrated fuzzy FUCOM and neutrosophic fuzzy MARCOS methodology. Science of The Total Environment, 788, 147763. (ii) Fuzzy Power Heronian function based CoCoSo method for the advantage prioritization of autonomous vehicles in real-time traffic management. Sustainable Cities and Society, 69, 102846.

Response 5: The literature review has been enriched with the studies you mentioned in new section "2.4. The Application of Quantitative Research in the Development and Regeneration of Industrial Base".

(Line 218-225) "According to the above research results, the novel extensions of the combined compromise solution methodology including the logarithmic method were proposed [22], in addition, a novel multi-criteria decision-making methodology based on Fuzzy Full Consistency Method and Neutrophilic Fuzzy Measurement Alternatives and Ranking were developed to establish the Compromise Solution Framework to improve the connectivity and convenience of road network in the industrial bases, which made the realization of the sustainable development of the industrial bases possible [23]."

Point 6: The authors need to discuss about the limitations of the proposed method as well as case study limitations, what are your recommendations for future works, how the proposed method solved the case study problem.

Response 6: We have added a new paragraph to discuss about the limitations of the method as well as case study limitations, and proposed method solved the case study problem.

(Line 535-248) "Although spatial syntax technology provides a visible and objective analysis method, there exist two limitations in the process of the simulation of spatial extension and road accessibility. Firstly, both Convex Map and Axial Map can only describe the two-dimensional relationship of the spatial morphology and ignore the consideration of the factors in three-dimensional space such as the height of buildings, ground elevation changes, and attraction of the color and noise of the space; Secondly, mandatory functions of block modules (schools, businesses, and hospital) and regulations and management of the traffic can influence people's behavior and vehicle flow along with getting rid of the passive influence of spatial morphology. The two limitations will result in the reduction of simulation analysis accuracy. In addition, In the 1950s, other regions of China also conducted large-scale industrial construction under the guidance of the Soviet Union's industrial planning patterns, such as the Southeast and Southwest. In order to better understand the practice level, we will continue our study to select more representative cases from other regions to implement a comparative study."

Point 7: How practitioners can use the proposed method in the real life problems, how the proposed method is useful for future studies.

Response 7: We have added new suggestion for practitioners to solve the real life problems , as well as discussed that how the proposed method is useful for future studies.

 (Line 551-568) "To improve the sustainability of the new emerging industrial cities, greater adoption of the planning pattern for the new industrial base of a single plant more than the new industrial base of a multi-plant joint type is encouraged, as it is easier to form a reasonably independent, circular, small satellite city, and it will not affect the expansion and development of urban spatial structures in the future. At the beginning of the planning, we suggest to determine the number and size of block modules and divide the living zone according to the functional requirements with different level. Firstly, the simulation of spatial extension and road accessibility along with data analysis should be conducted for several comparative planning plans, the land uses of block modules for different needs will be determined. Secondly, the road connectivity could be simulated based on the street grid established by the block module distribution. According to the above analysis results, the classes and width of the road could be determined. We even could determined the traffic rules to restrict the direction of traffic flow and better support the mandatory functions of block modules."

Best regards,

Yours sincerely

Rui Han (on behalf of all authors)

College of Art and Design, Creative Center for ArtSciArch, Jilin Jianzhu University, Changchun 130118, China.

Reviewer 2 Report

See attached

Author Response

Dear reviewer,

Thank you for handling the review of our manuscript entitled “A Comparative Study on Planning Patterns of Industrial Bases in Northeast China based on Spatial Syntax (Sustainability-1518321)”. In this revised version, we have carefully addressed all the issues raised by you. We sincerely appreciate your insightful comments and suggestions that greatly help improve our manuscript.

The following is a summary of the point-to-point response and revisions we have made to each of your comments. We look forward to hearing your more suggestion and the outcome of this latest version of the manuscript.

Point 1: The theoretical contribution is not clearly summarised in the introduction and conclusions sections. How do the findings specifically contribute to the existing literature, particularly in the research domains of sustainability?

Response 1: Thank you for your suggestion which benefits us a lot. We have rewritten the introduction and revised the conclusions to highlight the theoretical and technical contribution.

  1. Introduction

(Line 65-77) "The spatial morphology of industrial cities in Northeast China has been deeply influenced by the Soviet Union's industrial base planning theory and technology in the 1950s. From 1953 to 1957, under the industrial aid of the Soviet Union, a large number of giant industrial bases attaching to the main railways were built in the suburban areas of the original cities [1]. With the completion and running of these industrial bases, two positive changes emerged. On one hand, the spatial structure of the original cities has been expanded and continued; On the other hand, several independent satellite cities have been formed with their perfect functional configuration. After 70 years of development, the service functions of these industrial bases are constantly enriched and improved, along with constantly transforming and upgrading their production content and method. Nevertheless, due to the solidification and shackles of the original spatial structure and road network, the sustainable development of these industrial bases is facing great challenges [2, 3]."

(Line 78-89) "There are three main benefits to carrying out this study. First of all, the problem of the industrial base planning was deeply dissected with the scientific method, meanwhile, the internal logical relationship between road differentiation and spatial structure was revealed [4]; Secondly, Through quantitative study, the spatial function modules with high selectivity and low extension depth were clarified. Based on the above data, we proposed to modify and improve the spatial distribution of the functional modules in the industrial base, to optimize the flow direction of the road network, to eventually meet the needs of industrial production adjustment and future industrial regeneration development [5, 6]; Finally, this study can provide more early simulation analysis methods and strategies for the new industrial base planning in the future, which makes it more organically and harmoniously coexist with the original city while its sustainable development."

  1. Conclusions

(Line 535-548) "Although spatial syntax technology provides a visible and objective analysis method, there exist two limitations in the process of the simulation of spatial extension and road accessibility. Firstly, both Convex Map and Axial Map can only describe the two-dimensional relationship of the spatial morphology and ignore the consideration of the factors in three-dimensional space such as the height of buildings, ground elevation changes, and attraction of the color and noise of the space; Secondly, mandatory functions of block modules (schools, businesses, and hospital) and regulations and management of the traffic can influence people's behavior and vehicle flow along with getting rid of the passive influence of spatial morphology. The two limitations will result in the reduction of simulation analysis accuracy. In addition, In the 1950s, other regions of China also conducted large-scale industrial construction under the guidance of the Soviet Union's industrial planning patterns, such as the Southeast and Southwest. In order to better understand the practice level, we will continue our study to select more representative cases from other regions to implement a comparative study."

(Line 551-568) "To improve the sustainability of the new emerging industrial cities, greater adoption of the planning pattern for the new industrial base of a single plant more than the new industrial base of a multi-plant joint type is encouraged, as it is easier to form a reasonably independent, circular, small satellite city, and it will not affect the expansion and development of urban spatial structures in the future. At the beginning of the planning, we suggest to determine the number and size of block modules and divide the living zone according to the functional requirements with different level. Firstly, the simulation of spatial extension and road accessibility along with data analysis should be conducted for several comparative planning plans, the land uses of block modules for different needs will be determined. Secondly, the road connectivity could be simulated based on the street grid established by the block module distribution. According to the above analysis results, the classes and width of the road could be determined. We even could determined the traffic rules to restrict the direction of traffic flow and better support the mandatory functions of block modules."

Point 2: The literature review is too context specific. There are many previous work on space syntax and urban spatial structure which are not reviewed (suggest reading, for instance, https://nrl.northumbria.ac.uk/id/eprint/27794/ and doi.org/10.1177%2F0885412219853259) . The authors should summarize what is widely acknowledged and what remains to be explored. Also the literature review should be strengthened in a more critical way.

Response 2: We totally agree with your view. We came out a new section "2.4. The Application of Quantitative Research in the Development and Regeneration of Industrial Base" including the article you mentioned to discuss the previous work on spatial syntax and urban spatial structure .

(Line 203-225) "In the mid-1990s, with the adjustment of industrial production structure, more and more traditional industrial bases gradually lost their production functions and became industrial heritage. Their development and regeneration have been seriously focused in the field of urban planning. Since 2000, several novel mathematical models and statistical technology have played an increasingly important role in the process of industrial base regeneration planning, which symbolized that the research on their development and regeneration began to shift from qualitative study to quantitative study [17, 18]. The extension depth and connectivity of functional modules and street blocks of the industrial bases were objectively evaluated by applying spatial syntax theory and technology. The business plans and development strategies of cultural and creative industries were formulated based on the reorganization of the architectural spatial modules [19, 20]. The internal logical relationship between the road differentiation and the realization of block differentiation was better understood by deeply dissecting the relationship between the syntax of street networks and the differentiation of the size of the blocks in the industrial base. The historical evolution process and future development regulation of the industrial bases will be easily revealed [21]. According to the above research results, the novel extensions of the combined compromise solution methodology including the logarithmic method were proposed [22], in addition, a novel multi-criteria decision-making methodology based on Fuzzy Full Consistency Method and Neutrophilic Fuzzy Measurement Alternatives and Ranking were developed to establish the Compromise Solution Framework to improve the connectivity and convenience of road network in the industrial bases, which made the realization of the sustainable development of the industrial bases possible [23]."

Additionally, We came out with a new section "2.5. Summary " to make a summarize for the relevant literature review as well as strengthen it in a more critical way.

(Line 227-246) "The emergence, development, and maturity of the Soviet Union's industrial base planning pattern have its specific historical background, in which economy, practicality, ideology, and other factors permeate and couple with each other. The advantages of the planning pattern present that the land resources of production chain enterprises were efficiently integrated, a novel pattern of industrial clusters was formatted as the core, railway transportation functions were fully mobilized, the health environment and functions supporting living zone was improved. These advantages aim to create a novel socialist life pattern [24, 25]. However, the disadvantages are very obvious as follows: paying too much attention to the central axis planning pattern to highlight ideology, ignoring the rationality of road network, mixing too much subjective consciousness in the planning process, and resulting in weak support of the functional modules in the living zone. As a supporting part of the production zone, the location of the living zone did not take into account the impact of the wind environment, sunshine, and noise on the living quality for people, resulting in the deterioration of sustainable development. Over the years, the relevant studies have mainly focused on historical data research, value assessment research, theoretical and technical research on conservation and regeneration. There exists an extreme lack of quantitative research on the spatial morphology characteristics of the industrial base, which leads to the possibility of subjective judgment imbalance in its future regeneration strategies. The summary of relevant literature research in recent years is listed in Table 1."

(Line 247-248)

Table 1. The summary of relevant literature research in recent years.

Topic of Relevant Research

Main Viewpoints

Scholars and Publication Time

The planning pattern of the Soviet Union's industrial base

The historical background, the main contents and corresponding standards of the Soviet Union's industrial base planning pattern.

Binko, M.(1955), SCCSU(1975), Tan, Y.(1995), Li, Y.(2010), Sun, R.(2013), Niu, X.(2014), Jevremovic, L.(2014)

The transfer of the industrial base planning theory and technology from the Soviet Union to China

The localization process of American industrial technical assistance in the Soviet Union. The channel, content, and process of the transfer of the industrial base planning theory and technology from the Soviet Union to China. The inheritance and innovation of the planning theory and technology in China.

Shen, Z.(2002), Zhang, B.(2005), Zhang, Y.(2018), Wei, L.(2018),  Han, R.(2020),

Multi-value assessment and application of industrial bases in Northeast China

The qualitative research on historical value, cultural value, aesthetic value, scientific and technological value, and economic value. case application by establishing the Multi-value assessment system.

Dong, Z.(2004), Zhang, P.(2008), Wu, Y.(2018)

Regeneration and development of the   industrial bases

Development strategy and feasibility evaluation method of the regeneration of industrial heritage. Research on the theory and technology to serve the regeneration.

Guo, F.(2015), Yang, J.(2018), Han, R.(2020), Liu, X.(2019),   Li, L.(2021),

Development planning of the industrial bases based on quantitative research

The internal logical relationship between the road differentiation and the block differentiation. Algorithms for improving the connectivity of the road network.

Deveci,W.(2010), Lim, L.(2015), Liu, F.(2019), Pamucar,D.(2021)

(Line 248-256) " Since the 1950s, while absorbing the advanced planning technology of the Soviet Union, the industrial bases in Northeast China inevitably inherited many drawbacks, which leads to lots of problems for sustainable development of urban spatial morphology in the next 70 years. In order to improve the sustainability of these industrial bases, this study introduces spatial syntax technology to conduct quantitative analysis of functional modules and road axes, thus revealing the origin of the defects of the Soviet Union's industrial base planning pattern, establishing optimization strategies, and artificially improving the sustainability of the industrial bases by reorganizing roads flowing pattern and changing the land uses of functional modules."

Point 3: I would suggest the authors linking the policy implications to the research findings in a more concrete way. For instance, when the authors claim that “Changchun First Automobile Works Base is suitable to develop automobile industry tourism project with the subject of truck and car culture”, such arguments seem to have little connections to the analysis.

Response 3: We totally agree with your view. We have deleted the discussion content with less connections to the analysis.

(Line 584-590) Deleted Content " Harbin Three Power Industrial Base is suitable to develop  boiler generator industry tourism project with the subject of generator science and technology; Changchun First Automobile Works Base is suitable to develop automobile industry tourism project with the subject of truck and car culture; Shenyang Dadong Aeronautics and Astronautics Industrial Base is suitable to develop aerospace industry tourism project with the subject of aircraft manufacturing history in modern China."

Point 4: In addition, English language editing is suggested for this paper. I recommend the authors inviting a native speaker to proofread the manuscript before resubmission. For instance, it should be “indicates” in line 436.

Response 4: We have invited a native speaker to proofread the manuscript before resubmission (The English-Editing-Certification is in the attachment of this cover letter). We have corrected all grammar errors including the one you mentioned.

Best regards,

Yours sincerely

Rui Han (on behalf of all authors)

College of Art and Design, Creative Center for ArtSciArch, Jilin Jianzhu University, Changchun 130118, China.

Reviewer 3 Report

The manuscript is interesting and innovative in relation to the analyses of simulation through the software Convex map and Axial map, nevertheless I suggest to add the expressions and formulas of the synergy and the intelligibility  for understanding better the graphs at the page13. The definitions of the sinergy and intellegibility are insufficient for reconstructing the graphs.

The language can be improved , i.e. at the line 236 where the verb is missing.

Author Response

Dear reviewer,

Thank you for handling the review of our manuscript entitled “A Comparative Study on Planning Patterns of Industrial Bases in Northeast China based on Spatial Syntax (Sustainability-1518321)”. In this revised version, we have carefully addressed all the issues raised by you. We sincerely appreciate your insightful comments and suggestions that greatly help improve our manuscript.

The following is a summary of the point-to-point response and revisions we have made to each of your comments. We look forward to hearing your more suggestion and the outcome of this latest version of the manuscript.

Point 1: The manuscript is interesting and innovative in relation to the analyses of simulation through the software Convex map and Axial map, nevertheless I suggest to add the expressions and formulas of the synergy and the intelligibility  for understanding better the graphs at the page13. The definitions of the Synergy and Intelligibility are insufficient for reconstructing the graphs.

Response 1: Thank you for your suggestion which benefits us a lot. We have added the expressions and formulas of the synergy and the intelligibility for understanding better the graphs in Tables 5, 6, and 7.

 (Line 434-439) " The scatter chart on the right side of Table 5. presents the Correlation between the density of road network and Integration. The coupling relationship of the scatter chart conforms to the following equation:

                           Y = aX + b,                                  (1)

We obtained the data and the straight-line graph with the intelligent simulation of Depthmap Beta® 1.0 (Shenzhen University, team ARI. China)."

Point 2: The language can be improved , i.e. at the line 236 where the verb is missing.

Response 2: We have invited a native speaker to proofread the manuscript before resubmission (The English-Editing-Certification is in the attachment of this cover letter). We have corrected all grammar errors including the one you mentioned.

Best regards,

Yours sincerely

Rui Han (on behalf of all authors)

College of Art and Design, Creative Center for ArtSciArch, Jilin Jianzhu University, Changchun 130118, China.

Round 2

Reviewer 1 Report

All issues have been successfully addressed by authors. Just minor revision is required for Reference [22]. It will be "Deveci, M., Pamucar, D., & Gokasar, I. (2021). Fuzzy Power Heronian function based CoCoSo method for the advantage prioritization of autonomous vehicles in real-time traffic management. Sustainable Cities and Society, 69, 102846."

Reviewer 2 Report

All my concerns have been well addressed.

Reviewer 3 Report

The second version of the manuscript is greatly improved and enriched. Also the English language and style are better and more accurate.

The explanations of the analysis results are commented in more detail as well as the conclusions.

Last my suggestion, for future development of the manuscript, is the reading of the paper entitled: “M.G.D'Urso, D. Masi, G.Zuccaro, D. De Gregorio (2018) "Multicriteria Fuzzy Analysis for  a GIS -based Management of Earthquake Scenarios" COMPUTER AIDED CIVIL AND INFRASTRUCTURES ENGINEERING, 33 (2018) 165-179, doi: 10.1111/ mice.12335, that dealts with a multi-criteria fuzzy method for evaluating different alternatives in relation to the specific themes, i.e.  choices of industrial tourism projects, or, simply, for comparing the obtained results  with an other multi-criteria fuzzy approach.